# *Gluconacetobacter diazotrophicus* Changes The Molecular Mechanisms of Root Development in *Oryza sativa* L. Growing Under Water Stress

**DOI:** 10.3390/ijms21010333

**Published:** 2020-01-03

**Authors:** Renata Silva, Luanna Filgueiras, Bruna Santos, Mariana Coelho, Maria Silva, Germán Estrada-Bonilla, Marcia Vidal, José Ivo Baldani, Carlos Meneses

**Affiliations:** 1Departamento de Biologia–Centro de Ciências Biológicas e da Saúde / Programa de Pós-Graduação em Ciências Agrárias, Universidade Estadual da Paraíba, Rua Baraúnas, 351, Bairro Universitário, Campina Grande-PB 58429-500, Brazil; re.priscilaalmeida@gmail.com (R.S.); luannabeserra-uepb@hotmail.com (L.F.); brunacavalcantes23@gmail.com (B.S.); maryppgta@gmail.com (M.C.); silvamarialima@hotmail.com (M.S.); 2Corporación Colombiana de Investigación Agropecuaria—AGROSAVIA. Centro de Investigación Tibaitatá—Km 14 Vía Mosquera-Bogotá, Cundinamarca, Colombia; gaestrada@agrosavia.co; 3Embrapa Agrobiologia, Rodovia BR 465, km 07, Seropédica-RJ 23891-000, Brazil; marcia.vidal@embrapa.br (M.V.); ivo.baldani@embrapa.br (J.I.B.)

**Keywords:** Endophytic bacteria, red rice, effectiveness of root development

## Abstract

Background: Inoculation with *Gluconacetobacter diazotrophicus* has shown to influence root development in red rice plants, and more recently, the induced systemic tolerance (IST) response to drought was also demonstrated. The goal of this study was to evaluate the inoculation effect of *G. diazotrophicus* strain Pal5 on the amelioration of drought stress and root development in red rice (*Oryza sativa* L.). Methods: The experimental treatments consist of red rice plants inoculated with and without *strain* Pal5 in presence and absence of water restriction. Physiological, biochemical, and molecular analyses of plant roots were carried out, along with measurements of growth and biochemical components. Results: The plants showed a positive response to the bacterial inoculation, with root growth promotion and induction of tolerance to drought. An increase in the root area and higher levels of osmoprotectant solutes were observed in roots. Bacterial inoculation increased the drought tolerance and positively regulated certain root development genes against the water deficit in plants. Conclusion: *G. diazotrophicus* Pal5 strain inoculation favored red rice plants by promoting various root growth and developmental mechanisms against drought stress, enabling root development and improving biochemical composition.

## 1. Introduction

Roots are fundamentally important for growth and development, as they anchor the plant to its growth substrate, facilitate water and nutrient uptake from the soil, and sense and respond to environmental signals, such as abiotic stressors. Several studies indicate that water deficiency affects plants at physiological, biochemical, and molecular levels, by affecting cell division and expansion, root and leaf size, fresh and dry weight [1].

Plant growth-promoting bacteria (PGPB) are directly associated with plant roots and can exist within root tissues. Characterized by their plant-growth promoting properties, PGPB are a diverse group of bacteria that produce a wide range of enzymes and metabolites, influence nutrient acquisition, modulate hormone levels, and ameliorate the negative impact of abiotic stressors [2,3].

Development of drought tolerant genotypes has been the standard approach used to mitigate the problem of drought stress on rice, and conventional plant breeding techniques have indeed resulted in the development of high yielding, drought-tolerant varieties. The disadvantages of this approach are its time consuming and labor-intensive features and that it could lead to loss of other desirable traits from the plant gene pool [4]. Consequently, a microbial-based approach to mitigate drought stress can serve as a novel solution for improving the rice plant’s tolerance to restricted water availability. Although the roles of PGPB in growth promotion, nutrient management, and disease control are well known, their roles in the management of abiotic stress, such as drought, has only recently gained importance, as stated by Ngumbi and Kloepper [5].

*G. diazotrophicus* is considered an aero-tolerant endophytic diazotroph, for which oxygen is vital for the generation of large amounts of adenosine triphosphate (ATP) required for nitrogen fixation [6]. In rice (*Oryza sativa* L.), these bacteria may be responsible for providing up to 50–70% of the nitrogen demand, via the biological nitrogen fixation (BNF) process [7]. Such abilities, including carbohydrate metabolism, auxin biosynthesis, and gluconacin production, allow characterization of *G. diazotrophicus* as a PGPB. Plant growth substances, such as salicylic acid, auxins, gibberellins, cytokinins, and abscisic acid, are known to modulate plant responses against drought [8].

Here, we used red rice as the model plant species, considering that its role under drought conditions remains unclear and has been either marginally mentioned or entirely neglected. Roots are generally less fully analyzed than aboveground organs because they are difficult to observe, particularly in situ, and analysis methods tend to be laborious, imprecise, and difficult to standardize across experiments.

The occurrence of the plant–microorganism association under drought conditions may reveal an evolutionary interaction. Understanding these interactions in semi-arid environments could aid the development of new technologies, such as the use of inoculants, to mitigate stress and increase plant productivity. Here, we demonstrated that the inoculation of the *G. diazotrophicus* strain Pal5 improved root development in red rice grown under drought stress conditions, via physiological, biochemical, and real-time quantitative reverse transcription polymerase chain reaction (RT-qPCR) analysis of genes.

## 2. Results

To counteract the adverse effects of various abiotic stresses, plants have evolved complex mechanisms to improve survival, growth, and adaptation. *G. diazotrophicus* also regulates morpho-physiological, biochemical, and molecular responses in rice. Therefore, to analyze the effects of Pal5 on various biochemical parameters, red rice plants were harvested at different water availability levels.

### 2.1. Plant Development

Drought stress induces reduction in plant growth and development of rice due to the reduction in turgor pressure under stress; thus, cell growth is severely impaired. Drought affects both elongation and expansion growth and inhibits cell enlargement more than it does cell division. Here, the inoculation effect of *G. diazotrophicus* associated with different water restriction levels on growth of shoots, leaf areas, and roots is shown in Figure 1. There was a significant effect of the bacterial inoculation on growth of the plant shoot systems (Figure 1a,b). Inoculation alleviates suppression of shoot growth at 50% and 70% water availability compared to uninoculated plants (Figure 1a), while inoculation mitigates the leaf area at 30%, 50%, and 70% water availability compared to control plants (Figure 1b).

Concerning the shoot length, water restriction at 70% promotes higher growth, similar to inoculated plants at 100% field capacity. The growth of inoculated plants at 70% water restriction level did not statistically differ from inoculated plants grown at 100% field capacity (control).

*G. diazotrophicus* ameliorates the root growth at water restriction levels of 50% and 70%, which statistically differ from the inoculated plants grown at 100% field capacity (Figure 1c,d). The other treatments showed a similar pattern to that observed for root length and lateral root. 

### 2.2. Gas Exchange

Knowledge about physiological responses of rice growing under drought conditions may contribute to the ongoing studies on drought tolerance in rice. It is known that drought stress affects various physiological processes and induces several physiological responses in plants, which help plants to adapt to the limiting environmental conditions surrounding them. Optimization of these physiological processes is a pre-requisite for increasing productivity under water stress.

Analysis of the gas exchange showed that inoculation with *G. diazotrophicus* promoted higher water-use efficiency (ability of the crop to produce biomass per unit of water transpired) (Figure 2a), while the instantaneous efficiency of carboxylation was closely related to the intracellular CO_2_ concentration and CO_2_ assimilation rate (Figure 2b). Rehydration of the plants increased gas exchange either in presence or absence of inoculated bacteria, independent on the initial water restriction values. On the other hand, rehydration increased the levels of water-use efficiency and instantaneous efficiency of carboxylation to values of the inoculated plants growing at 100% field capacity.

Water-use efficiency in red rice plants with and without inoculation with *G. diazotrophicus* tended to decrease as the water percentage levels increased in the soil. Under full irrigation, no significant difference was observed between treatments. Regarding to the other water restriction levels (70%, 50%, and 30%), there was a significant decrease in the water-use efficiency compared to the control while a smaller increase was observed for the water-use efficiency, mainly at 30% and 70% for uninoculated plants.

A significant increase in water-use efficiency was noted in all treatments one hour after replenishing irrigation, suggesting that inoculated plants have a more efficient stomatal control, which may promote greater adaptability to water deficit conditions.

The instantaneous carboxylation efficiency decreased as the water deficit increased either in inoculated or uninoculated red rice plants. After rehydration, soil water percentage levels of 70%, 50%, and 30% showed a large increase in instantaneous carboxylation efficiency, however, they did not statistically differ from 100%, which continued showing the same rate. Analyzing the percentages of water in the soil of inoculated and uninoculated red rice, all levels presented lower values than the control plants (without water restriction), therefore showing significant differences among them, with the inoculated plants responding better to 70% in contrast to 30%, that showed the lowest values similar to the uninoculated treatment. After rehydration, there was a rapid increase in carboxylation efficiency at all water restriction levels, except for 100%, which showed the same amount for all rehydrated levels.

### 2.3. Chlorophyll Fluorescence

Photosynthesis is the main metabolic process determining crop production and is affected by drought stress. An initial fluorescence (F0) value is a measurement of the steady-state fluorescence signal, which is correlated with all "open" reaction centers and refers to the fluorescence emission by chlorophyll a molecule from the photosystem II (FSII) light-collecting complex. To determine the impact of drought and inoculation on F0, we measured this parameter in inoculated and uninoculated plants growing at different water availability levels. Results from the inoculated plants showed significant differences among the levels of 70%, 50%, and 30% soil water. However, this effect was not observed in the control plants (100%) that presented no difference among the treatments. Plant rehydration showed an increase in F0 fluorescence in relation to the stress period, but there was no significant difference at the 100% level (Figure 3a).

The variable fluorescence/ maximum fluorescence (Fv/Fm) ratio or FSII maximum quantum yield (which is a measurement ratio that represents the maximum potential quantum efficiency of FSII if all capable reaction centers were open) showed significant differences among soil water percentage levels, mainly at 70%, 50%, and 30% of field capacity, as well as between inoculated and uninoculated plants, except for those at 100% field capacity (Figure 3b). Rehydration showed an increase in the Fv/Fm ratio in all soil water percentage levels compared to water stress treatments with the same behavior at all water restriction levels.

### 2.4. Photosynthetic Pigments

Drought causes many changes that alter metabolic functions, and one of these changes is either loss or reduction of the photosynthetic pigment synthesis. This event results in a decline in light harvesting and the generation of reducing power, which are a source of energy for the dark reactions of photosynthesis. These changes in the amount of photosynthetic pigments are closely associated to plant biomass and yield. Here, the inoculation of *G. diazotrophicus* on red rice plants promoted significant differences in the chlorophyll a, chlorophyll b, total chlorophyll, and carotenoid content at all soil water percentage levels (Figure 4a–d). The drought stress negatively impacted the level of pigments regardless of bacterial inoculation; nevertheless, under these conditions of water limitation, inoculation always improved the content of photosynthetic pigments. Red rice plants grown without water restriction obtained higher photosynthetic pigment production, while a decrease of the soil water percentages to 70%, 50%, and 30% showed lower pigment concentrations.

### 2.5. Phytohormones and Osmoprotective Solutes

Phytohormones are growth regulators and are involved in stress tolerance. Several studies have confirmed their role in mediating plant responses against drought stress conditions through a series of signal transduction pathways. As water deficit occurs, plants accumulate different types of organic and inorganic solutes in the cytosol to lower osmotic potential, thereby maintaining cell turgor (i.e., trehalose and α-tocopherol). This biochemical process is known as osmotic adjustment, which strongly depends on the rate of plant water stress.

HPLC-based phytohormone profiles of plant extracts revealed that all water deficiency levels (except 30%) produced indole-3-acetic acid (IAA), gibberellic acid 3 (GA_3_), gibberellic acid 1 (GA_1_), and cytokines (Cyt) in relatively high amounts compared to the control (Table 1a and b). Overall, a decrease in water availability and inoculation was positively correlated with an increase in hormone concentrations. This suggests that hormones are secreted as part of the stress response and that bacterial inoculation can directly and/or indirectly affect hormone plant levels. To confirm this phenomenon, we measured the expression patterns of several key genes related to hormones in both partners (Figure 5).

RT-qPCR analysis of the plant and bacterial materials revealed significant differences in the gene expression of indole-3-acetic acid (IAA), gibberellic acid 3 (GA_3_), gibberelic acid 1 (GA_1_), and cytokines (Cyt) during the water restriction period (15 days) and after rehydration.

The highest differential expression (in all genes analyzed) was observed in red rice plants inoculated with the strain Pal5 and subjected to 70% of total water availability (Figure 5a). The relative expression of IAA, GA, and Cyt genes increased ~60.3-, 39.2-, and 35.1- fold, respectively, relative to the uninoculated plants cultivated at field capacity.

In case of the *G. diazotrophicus* Pal5 strain, the highest differential gene expression (in all genes analyzed) was observed when the plants were also subjected to 70% of total water availability. The relative expression of the IAA, GA, and Cyt genes increased ~44.9-, 25.1-, and 23.4- fold, respectively, relative to the bacteria in plants cultivated at the field capacity (Figure 5b).

Induced Systemic Tolerance (IST)-inducing *G. diazotrophicus* can be a mechanism to reduce the damage caused by adverse conditions, since it increases the abiotic stress tolerance capacity of plants. Filgueiras et al. [3] showed that bacteria N-acyl homoserine lactones (AHL) is also effective in systemic induction of drought stress tolerance in red rice. Therefore, AHL can be a potential candidate for a novel general elicitor for plant defense, as it is produced in red rice in effective amounts, induces salicylic acid (SA) and jasmonic acid (JA) (pathogenesis related (PR) genes) systemically, leads to enhanced gene expression for typical defense-related proteins, and consequently increases tolerance against drought stress.

Mean values were calculated from the results of three replicates, with standard errors. Averages followed by identical capital letters do not differ in the inoculation condition for each soil water treatment, and averages followed by identical lowercase letters do not differ in the percentage of soil water.

We noted that trehalose and α-tocopherol levels increased among the treatments with water restriction (70%, 50%, and 30%), mainly in the inoculated plants (Table 1). On the other hand, there was no significative difference between the trehalose levels of uninoculated rice plants grown at 100% (positive control) and 50% field capacity. There were also no significant differences among treatments after rehydration of rice plants. A similar behavior was observed for the α-tocopherol accumulation, for which uninoculated and rehydrated rice plants maintained similar values, while increments were observed in the inoculated plants exposed to water restriction conditions.

### 2.6. RT-qPCR Analysis of Root Growth and Developmental Genes

At the molecular level, the response to drought stress is multigenic. Through high-throughput molecular studies, several genes that respond to drought stress at the transcriptional level have been reported [3]. In this study, we described the expression of genes and regulatory pathways involved in the development of root systems of rice *(Oryza sativa* L.) [9], including crown roots, lateral roots, root hairs, and root length (Table 2). Two genes showed decreased expression levels, whereas 21 were overexpressed, according to the RT-qPCR analysis. All genes had their protein products classified according to clusters of orthologous groups (COG). The overexpressed COG analysis showed that the most representative genes were related to “root elongation”, “cell division and elongation”, “crown root initiation”, and “lateral root emergence”. The main under-expressed COG classes included “lateral root initiation”.

The RT-qPCR analysis of the red rice root plants revealed significant differences in gene expression of root development modulating genes during the water restriction period (15 days). The highest differential expression (in all genes analyzed) was observed in red rice plants inoculated with Pal5 strain and subjected to 50% total water availability. Relative gene expression increased 10–50 fold compared to uninoculated field-grown plants (Table 2). Interestingly, after 1 h rehydration, the gene expression tended to decrease to a level equivalent to that of the uninoculated control.

*G. diazotrophicus* presented positive effects in red rice (Figure 6), such as tolerance to drought, increments in shoot and root growth, improvement of physiological and biochemical attributes, and increased gene expression for phytohormone production and root growth and development.

The principal component analysis (PCA) showed a tendency of clustering the data in two groups. The gene expression of two auxin signaling pathways formed a very distinct group apart from the other parameters. This may be due to the specific characteristic of the genes located in the same quadrant. OsIAA11 and OsIAA13 (auxin signaling) were reported to regulate lateral roots (LR) formation, as LR development is inhibited in the gain-of-function mutants. Osiaa11 and Osiaa13; however, other root components, including crown roots and root hairs, are affected [9].

The results indicate that *G. diazotrophicus* inoculation enhances root and shoot growth, induced mainly from morphological and physiological changes in inoculated plant roots and enhancements in water and plant nutrient uptake. Plants exposed to stress conditions responded by increasing both osmoprotectant solutes and gene expression levels, which may help the plant to survive under adverse conditions.

## 3. Discussion

Sustainable rice production is currently confronting issues such as declining soil fertility, water scarcity, and degradation of the environment. Of these, water deficit is one of the foremost problems encountered during rice production, limiting crop productivity. Improving water deficit tolerance in rice through sustainable environment-friendly strategies is the key to deliver food security to the continuously rising population.

The development of the optimal leaf area is important for photosynthesis and dry matter yield. Inoculation of *G. diazotrophicus* promoted maximum increase in leaf area either under unstressed or drought conditions. *Pseudomonas* spp. are also known to increase total microbial activity, shoot and root lengths, and total dry weight of wild plants [10]. The inoculation of exopolysaccharide-producing bacteria also promoted the development of much more root systems, which subsequently increase the shoot growth [11]. In our study, all treatments have increased shoot length in both drought and unstressed plants than uninoculated controls. Inoculation of *G. diazotrophicus* shows a maximum increase in shoot length. Inoculation of *G. diazotrophicus* also increases total shoot length of maize, as demonstrated by Sandhya et al. [12].

These phenotypic differences in tolerance of plants to drought stress were associated with changes in the root system architecture, although there were some differences between hosts in response to the *G. diazotrophicus* strain Pal5 (Figure 7). For instance, one significant effect on the root system architecture is that compared to uninoculated plants, more branched roots in red rice were produced. The production of root length, surface area, and more root tips has been previously correlated with water stress tolerance and overall improvements in maintaining plant productivity under drought conditions [13]. Root length and surface area contribute to better soil exploration, whereas the proliferation of higher-order roots that result in more root tips are important for root water uptake capacity [14]. Previous research demonstrated that reduction in root diameter may enable relatively higher growth rates and rapid resource acquisition through expansion of the root system coupled with lower investment in dry biomass [15].

These results suggest that induction of greater root length may be related to the production of IAA and/or other yet unknown metabolites produced by PGPB [16]. The benefits of inoculation on root length and plant growth may possibly involve the proximity of the site of interaction between PGPB and plant tissues [17].

We noted that water-use efficiency and instantaneous efficiency of carboxylation were negatively affected by water deficiency, both in uninoculated and inoculated red rice plants. However, colonization with *G. diazotrophicus* positively modulated the response of stressed plants to gas exchange. In cases of low water availability in the soil, stomatal closure occurs to avoid the loss of water by transpiration, via changes in turgor pressure of the guard cells or by the action of ABA. In this sense, the endophytic bacteria *G. diazotrophicus* Pal5 may have altered the ABA levels in the stomata, causing a decrease in ABA concentration or an increase in the concentration of cytokinin produced, thereby ensuring the maintenance of stomatal opening [18].

Additionally, it is observed that FSII operating efficiency increased in inoculated plants. This parameter measures the proportion of light absorbed by chlorophyll molecules associated with FSII [19]. Red rice plants inoculated with *G. diazotrophicus*, therefore, showed an increase in overall photosynthetic capacity in vivo. Zhang et al. [20] showed that certain PGPB elevate photosynthesis in *Arabidopsis* through the modulation of endogenous sugar/abscisic acid (ABA) signaling and establish a regulatory role for soil symbionts in plant energy acquisition. Some PGPB are known to possess volatile organic compounds that function as modulators, sensing primary and secondary metabolism, stress responses and hormone regulation mechanisms in plants [20]. An Fv/Fm value of around 0.83 is an optimal measurement for most plant species [19]. This suggests that *G. diazotrophicus* promotes a healthy FSII in inoculated plants, and other authors have obtained similar results [21] of increased Fv/Fm values when using PGPB.

The results of the present study show that drought stress reduces the content of photosynthetic pigments, such as the contents of chlorophyll and carotenoids, in red rice plants. It has been observed that drought stress hinders photosynthetic machinery of rice plants by altering the levels of chlorophyll a, chlorophyll b, carotenoids, and net photosynthetic activity [22]. Enhancement in photosynthetic pigments could occur in the presence of bacteria that promote plant growth, ultimately by increasing nutrient uptake in plants through phosphate solubilization and by the exudation of essential substances that play a crucial role in the synthesis of the photosynthetic pigments required for light harvesting complexes and their photoassimilation [23]. Inoculation of *G. diazotrophicus* in red rice enhanced the levels of chlorophyll under water deficit. It was shown by Rizvi et al. that *Azotobacter chrococcum* enhanced the chlorophyll contents of *Zea mays* plants when exposed to abiotic stress [24].

Phytohormones produced by bacteria provide possible ways of augmenting plant growth under unfavorable environmental conditions [25]. These phytohormones relieve plants from abiotic stressors and improve survival rates [26,27]. There is an array of phytohormones, such as cytokinins, auxins, gibberellins, ethylene, abscisic acid, and jasmonates, which either promote shoot growth or regulate growth-inhibitory processes, such as dormancy, abscission, and senescence, thereby controlling growth activities in plants [28,29].

Trehalose and α-tocopherol are indicator molecules of drought stress. They accumulate in plant tissues under abiotic stress, serving as osmoprotectant molecules [5]. In our study, trehalose and α-tocopherol increased significantly in the leaves of all inoculated plants. The best performance was observed for plants inoculated with *G. diazotrophicus* that showed the highest trehalose and α-tocopherol content in roots. These results suggest a protective effect of the inoculated bacteria on the plants, resulting in a greater accumulation of these solutes in the tissues.

Significant progress has been made in our understanding of the genetic control of root development in rice, particularly through RT-qPCR analysis of root development. Though many genes involved in root development have been identified (Table A1), our knowledge about the molecular mechanisms of root elongation, crown root development, LR development, and root hair formation is still fragmented. Most of the genes related to root elongation, cell division and elongation, crown root initiation, and lateral root emergence had showed increased expression when plants were inoculated with *G. diazotrophicus*; however only two genes that control the lateral root initiation showed decreased expression. Our results indicate that *G. diazotrophicus* is capable to modulate root growth and development, thus allowing the mitigation of adverse conditions. Therefore, IST-inducing *G. diazotrophicus* can be useful in reducing damage caused by adverse conditions, by increasing the abiotic stress tolerance capacity of plants [3].

## 4. Materials and Methods

### 4.1. Growth Conditions and Strains

The *G. diazotrophicus* strain Pal5 (BR 11281, ATCC49037) was provided by Johanna Döbereiner at the Biological Research Centre (Embrapa Agrobiologia, Seropédica, Rio de Janeiro, Brazil). The bacteria were recovered from stocks after growing in an LGI-P liquid medium [30]. The LGI-P medium (semisolid and solid) was used to evaluate the nitrogen-fixing ability, purity, and rice colonization ability of the *G. diazotrophicus* Pal5 strain [30]. In the case of the liquid culture, the LGI-P medium was supplemented with 10 mM NH_4_(SO_4_)_2_ [31]. Incubation was carried out in the dark at a temperature of 28 °C for 4 to 7 days.

For the field experiment, red rice seeds were first peeled and disinfected, as previously described by Hurek et al. [32]. The seeds were germinated for 4 days at 28 °C in the dark on agar plates (1% agar) containing 10 times diluted LB medium. Seedlings without visible contamination were used to establish the experiment. A portion of the seedlings were macerated in saline solution and inoculated (100 µL) on LB, Dygs, LGI, and LGI-P agar plates, as well as into N-free semisolid JMV, JNFb, and LGI-P media [33], to check for the presence of nitrogen-fixing culturable microorganisms.

### 4.2. Experimental Design and Treatments

The experiment was performed in greenhouse conditions, using modified lysimeters containing sterilized substrates, and comprising a mixture of sand, pulverized coal (3:1, *v*/*v*), and sterile distilled water. Lysimeters were arranged in a completely randomized design with eight treatments and five replicates. 

The experiment was carried out on the red rice genotype (405 Embrapa Meio Norte) with and without seed inoculation, using the *G. diazotrophicus* Pal5 strain by which samples were subjected to four water restriction conditions: 30–35%, 50–55%, 70–75%, and 100% field capacity. The uninoculated plants (control treatment) were maintained in soil at full field capacity (100% water availability). The experiment was carried in a completely randomized design with eight treatments and five replicates, wherein each plot consisted of a lysimeter containing 60 plants. The red rice seeds were inoculated by a peat coating with a 10^8^ CFU/mL bacterial suspension and planted in the lysimeter 48 h later.

Irrigation was carried out along with the sowing, with the soil moisture maintained near field capacity. When the plants reached phase R3 (panicle emission), they were subjected to the water restriction regime (for 15 days). Soil samples were collected daily, and soil water potential was measured psychrometrically by the dew point potential meter WP4-T (Decagon Devices Inc., Pullman, WA, USA) using the water retention equation 1:y = 0.0492^−0.32^(1)

This procedure determine the amount of water replenished per day at each initial level of water restriction, until the grain filling, when larger amounts of R6 (milled grains) elongated grains were presented. After the water restriction regime (15 days), the plants were irrigated normally and analyzed 1 h after rehydration to verify possible normality in their biochemical and molecular apparatus.

Estimation of the population of inoculated bacteria that colonized the roots and leaves of the red rice plants was performed. According to Filgueiras et al. [3], the population of *G. diazotrophicus* Pal5 was higher than 10^5^ CFU/g per tissues in the roots and leaves of all inoculated treatments. In contrast, no bacteria were detected on the LGI-P medium for the uninoculated plants.

### 4.3. Analysis of Agronomical, Physiological, Biochemical, and Molecular Parameters

#### 4.3.1. Plant Morphological Parameters

Plant height was measured from randomly selected hills, which involved selecting main shoots and recording plant height from the ground level to the base of the fully opened leaf. The mean plant height was assessed and expressed in cm.

Total leaf surface area was determined by scanning all green leaves of selected plants using a stationary leaf area meter LI-3100C (LI-COR, Lincoln, NE, USA). Root lengths were obtained by scanning whole root systems of individual plants (30 plants treatment^−1^) and analyzing the images using WinRhizo® software (Regent Instrument Inc., Quebec, QC, Canada).

#### 4.3.2. Gas Exchange and Chlorophyll Fluorescence

The red rice plants were evaluated for physiological establishment in relation to drought stress, and the water-use efficiency (liquid photosynthesis / transpiration) [(µmol m^−2^ s^−1^) (µmol de H_2_O m^−2^ s^−1^)^-1^] and instantaneous efficiency of carboxylation (liquid photosynthesis/internal carbon concentration) [(µmol m^−2^ s^−1^).(µmol m^−2^ s^−1^)] were measured [34]. The measurement was done from the apical leaf, using the portable infrared gas analyzer (IRGA) model LCPro (ADC, Hoddesdon, UK) with an airflow of 300 mL/min and a coupled light source of 995 μmol m^−2^ s^−1^ rating was used.

In the same leaves, of which gas exchange processes were analyzed, foliar tweezers were placed, and after a period of 30 min of dark adaptation [34], the fluorescence parameters, such as the: Initial fluorescence (F0) and the quantum efficiency of the photosystem II (Fv/Fm), were determined using the pulse-modulated fluorometer OS5p (Opti-Sciences, Hudson, NH, USA).

#### 4.3.3. Plant Biochemical Parameters

Photosynthetic pigments (chlorophyll a, chlorophyll b, and carotenoids) were extracted from the needles for photosynthetic measurements. Pigment analysis was performed according to the methods described by Wellburn [35]. Ten milligrams (FW) of needles was extracted with 2 mL of dimethyl sulfoxide for 12 h in the dark at 45 °C. Total chlorophyll, chlorophyll a, chlorophyll b, and carotenoid contents were calculated at absorbances of 665, 649 and 480 nm. The colored compound was measured using a Multiskan GO plate reader (Thermo Fisher Scientific; Waltham, MA, USA).

Auxin, gibberellin (GA_1_ and GA_3_), cytokinin, trehalose, and α-tocopherol levels were determined using high-performance liquid chromatography (HPLC) (Thermo Fisher Scientific; Waltham, MA, USA). Approximately 50–100 mg of fresh red rice roots was sealed in 1.5 mL snap-cap vials. After being frozen in liquid nitrogen, the roots were ground into powder, and 500 µL of 1-propanol/H_2_O/concentrated HCl (2:1:0.002, vol/vol/vol) with internal standards (10–50 ng) were added, followed by agitation for 30 min at 4 °C. Then, CH_2_Cl_2_ (1 mL) was added, followed by agitation for another 30 min and centrifugation at 13,000g for 5 min. After centrifugation, two phases were formed, and plant debris was in the middle of the two layers. The lower layer (around 1 mL) was concentrated and re-solubilized in MeOH (0.3 mL), of which 25 µL was injected to the column for analysis. The lower layer (25 µL) was also directly injected to column for analysis.

Auxin, gibberellin (GA_1_ and GA_3_), cytokinin, trehalose, and α-tocopherol were separated by an HPLC equipped with a reversed-phase column C18 Gemini 5µ, 150 × 2.00 mm (Phenomenex, Torrance, CA, USA) using a binary solvent system composed of water with 0.1% HCO_2_H (A) and MeOH, with 0.1% HCO_2_H (B) as a mobile phase at a flow rate of 0.3 mL.min^−1^. Separations were performed using a gradient of increasing MeOH content. The initial gradient of methanol was maintained at 30% for 2 min and increased linearly to 100% at 20 min [36].

#### 4.3.4. Plant Molecular Parameters

Roots with and without inoculation, as described above, were gently removed from the lysimeter and immediately immersed in 20 mL of cold disinfectant solution (95% ethanol and 5% phenol-pH 4.0) for 5 min, followed by immersion in 50 mL of distilled cold water for 3 min, and vortexing (30 s) to remove the bacteria adhering to the root surface. The samples were frozen in liquid nitrogen, sliced, and then transferred to 1.5 mL microtubes containing a mixture of ~250 mg glass and silica beads (both with 1 mm diameter) (Thermo Fisher Scientific; Waltham, MA, USA). Total RNA was extracted according to the bead/SDS/phenol method described by Holmes et al. [37]. The concentration, and integrity and purity of the sample were determined with a Qubit 4.0 Fluorometer (Thermo Fisher Scientific; Waltham, MA, USA) and an L-Quant spectrophotometer (Loccus, São Paulo, SP, Brazil), respectively.

After DNase I treatment, 1 μg RNA was used for cDNA synthesis using a SuperScript® VILO™ kit (Thermo Fisher Scientific; Waltham, MA, USA). The RT-qPCR reactions were performed in a StepOnePlus Real-Time PCR thermocycler (Thermo Fisher Scientific; Waltham, MA, USA). Each reaction contained 12.5 µL of Power SYBR Green PCR Master Mix (Thermo Fisher Scientific; Waltham, MA, USA), 0.4 µL (1 mM) of each specific oligonucleotide, appropriate amounts of cDNA (1 µL, 1:4 dilution), and 10.7 µL of PCR water, to yield a final volume of 25 µL. The RT-qPCR reactions were carried out under the following conditions: initial denaturation at 95 °C for 3 min followed by 40 cycles of 95 °C for 30 s and 60 °C for 30 s, and a final extension at 60 °C for 5 min. The 2^−ΔΔCt^ method [38] was used for the relative quantification, and 23S rRNA was used as the bacterial endogenous control, while ubiquitin was used as the plant endogenous control. The PCR primers used for all the tested genes are listed in supplementary information of table A1 and A2. The quantity of target genes was normalized to the quantity of the endogenous control for each condition. The experiments were carried out in biological triplicates (each with two technical replicates).

#### 4.3.5. Statistical Analysis

Data are expressed as means ± SEM and were analyzed by two-way statistical analysis of variance (ANOVA) followed by a Tukey’s test. All treatments were performed in triplicate. Data analyses were carried out using SigmaPlot 11.0 software (Systat Software, San Jose, CA, USA). In all cases, the differences were considered significant at *p* < 0.05. The principal components analysis (PCA) was determined using R software (R Development Core Team, St. Louis, MO, USA).

## 5. Conclusions

Drought stress is a severe environmental constraint to agricultural productivity. *G. diazotrophicus* plays an important role in conferring tolerance and adaptation of red rice plants to drought stressors and has the potential role in solving future food security issues. The mechanisms elicited by *G. diazotrophicus*, such as triggering osmotic responses and inducing novel genes, play a vital role in ensuring red rice plant survival under drought stress. The root development of drought tolerant crop varieties through genetic engineering and plant breeding is essential and yet a long-drawn process, whereas *G. diazotrophicus* inoculation to alleviate drought stressors in plants opens a new opportunity and perspective on the application of microorganisms in dry land agriculture.

## Figures and Tables

**Figure 1 ijms-21-00333-f001:**
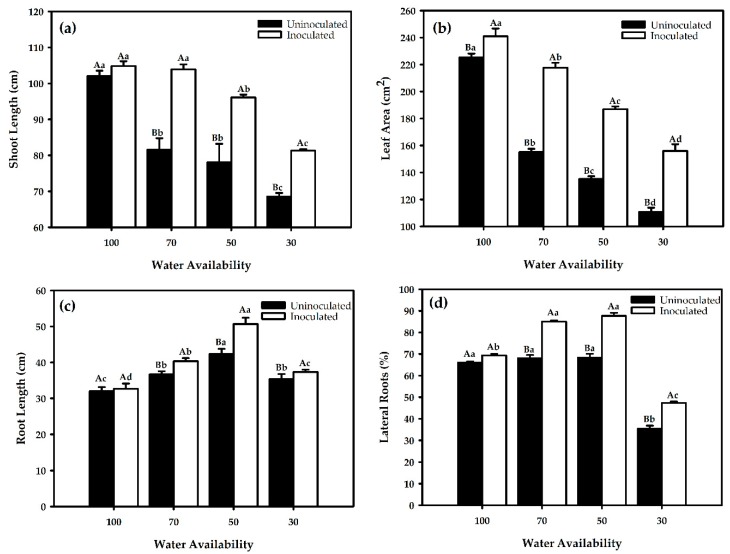
Shoot length (**a**), leaf area (**b**), root length (**c**), and lateral roots (**d**) in red rice under different water restriction and inoculation conditions. Averages followed by identical capital letters do not differ in the inoculation condition for each soil water treatment, and averages followed by identical lowercase letters do not differ in the percentage of soil water. Bars represent the standard error (n = 3).

**Figure 2 ijms-21-00333-f002:**
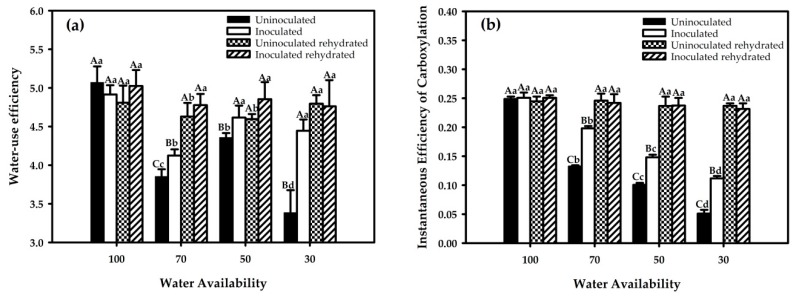
Gas exchange in red rice under different water restriction and inoculation conditions. Shown is of water-use efficiency (**a**) and instantaneous efficiency of carboxylation (**b**). Averages followed by identical capital letters do not differ in the inoculation condition for each soil water treatment, and averages followed by identical lowercase letters do not differ in the percentage of soil water. Bars represent the standard error (n = 3).

**Figure 3 ijms-21-00333-f003:**
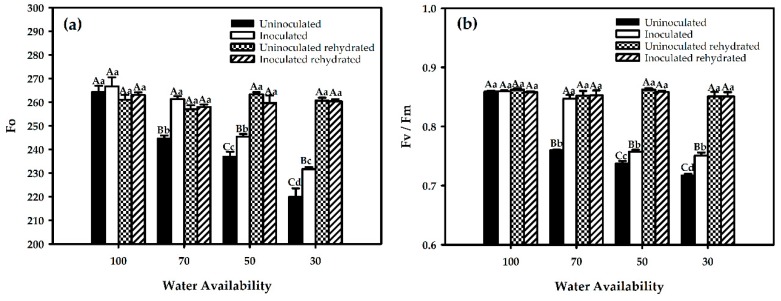
Initial fluorescence (**a**) and FSII maximum quantum yield (**b**) in red rice under different water restriction and inoculation conditions. Averages followed by identical capital letters do not differ in the inoculation condition for each soil water treatment, and averages followed by identical lowercase letters do not differ in the percentage of soil water. Bars represent the standard error (n = 3).

**Figure 4 ijms-21-00333-f004:**
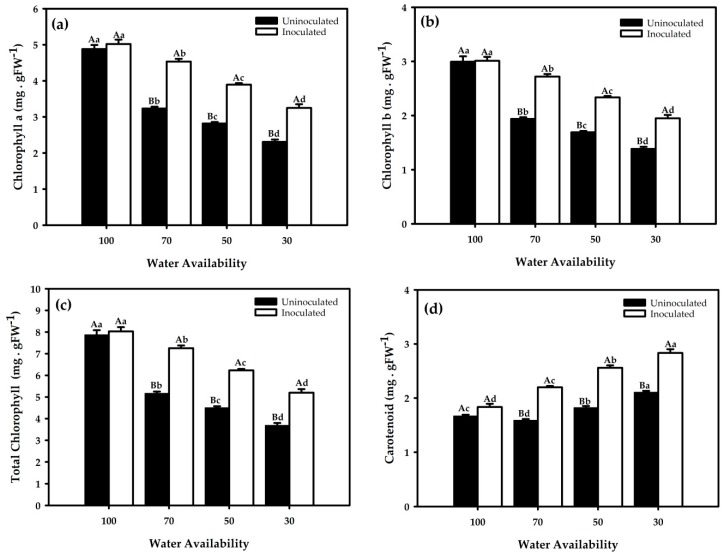
Photosynthetic pigment content. Chlorophyll a (**a**), chlorophyll b (**b**), total chlorophyll (**c**), and carotenoids (**d**) in red rice under different water restriction and inoculation conditions. Averages followed by identical capital letters do not differ in the inoculation condition for each soil water treatment, and averages followed by identical lowercase letters do not differ in the percentage of soil water. Bars represent the standard error (n = 3).

**Figure 5 ijms-21-00333-f005:**
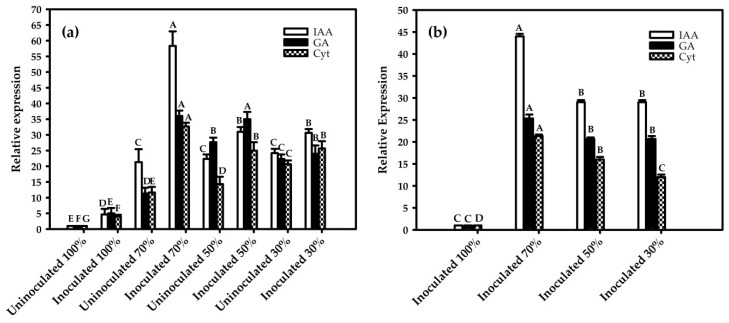
RT-qPCR analysis of phytohormone key genes during drought stress. (**a**) Red rice genes and (**b**) *G. diazotrophicus* genes. Averages followed by identical capital letters do not differ in the inoculation condition for each soil water treatment. Bars represent the standard error (n = 3).

**Figure 6 ijms-21-00333-f006:**
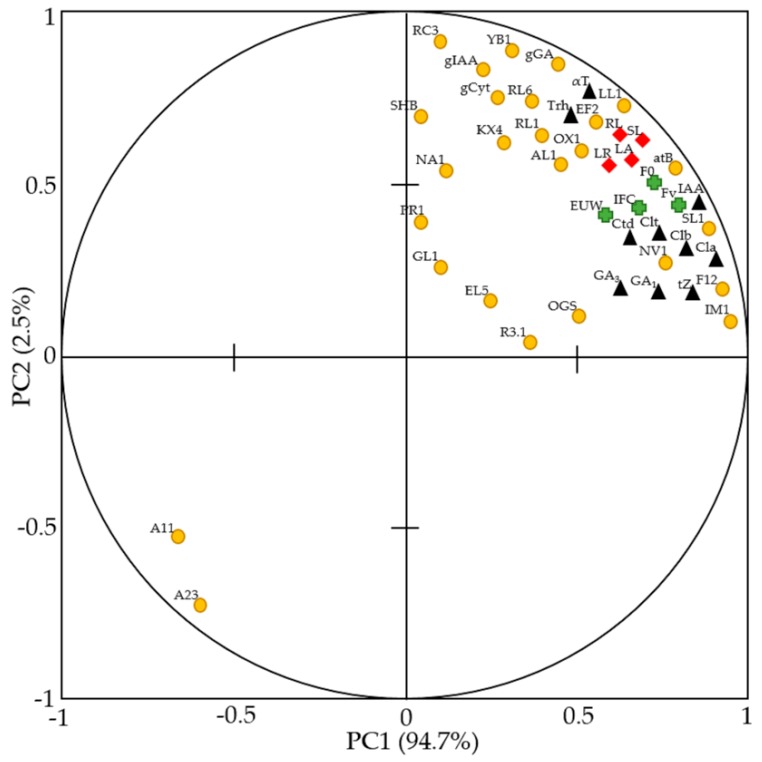
Principal components analysis (PCA) between the shoot and root growth reveals improvement of physiological and biochemical attributes and increased gene expression for phytohormone production and root growth and development. Data are from evaluations performed at 15 days after stress, for red rice plants inoculated with *G. diazotrophicus* and grown under water stress (50% field capacity. Symbols in red represent the growth parameters, in green represent the physiological parameters, in black represent the parameters of biochemists, and in yellow represent the parameters of gene expression. SL: shoot length; LA: leaf area; RL: root length; LR: lateral roots; WUE: water-use efficiency; IFC: instantaneous efficiency of carboxylation; F0: initial fluorescence; Fv: FSII maximum quantum yield; Cla: chlorophyll a; Clb: chlorophyll b; Clt: total chlorophyll; Ctd: carotenoids; IAA: indole-3-acetic acid; GA_1_: gibberellic acid 1; GA_3_: gibberellic acid 3; tZ: trans-zeatin; Trh: trehalose; αT: α-tocopherol; NA1: OsGNA1; OGS: OsMOGS; GL1: OsDGL1; NV1: OsCYT-INV1; R3.1: OsGLR3.1; atB: OsGatB; SL1: OsASL1; EL5: OsEL5; PR1: OsSPR1; YB1: OsMYB1; F12: OsARF12; RF2: OsERF2; IM1: OsAIM1; SHB: SHB; RL1: ARL1/CRL1; A23: OsIAA23; AL1: OsNAL1; RL6: OsCHR4/CRL6; KX4: OsCKX4; A11: OsIAA11; LL1: OsSLL1; RC3: OsORC3; OX1: OsHOX1; gIAA: auxin biosynthesis; gGA: GA biosynthesis; and gCyt: cytokinin biosynthesis.

**Figure 7 ijms-21-00333-f007:**
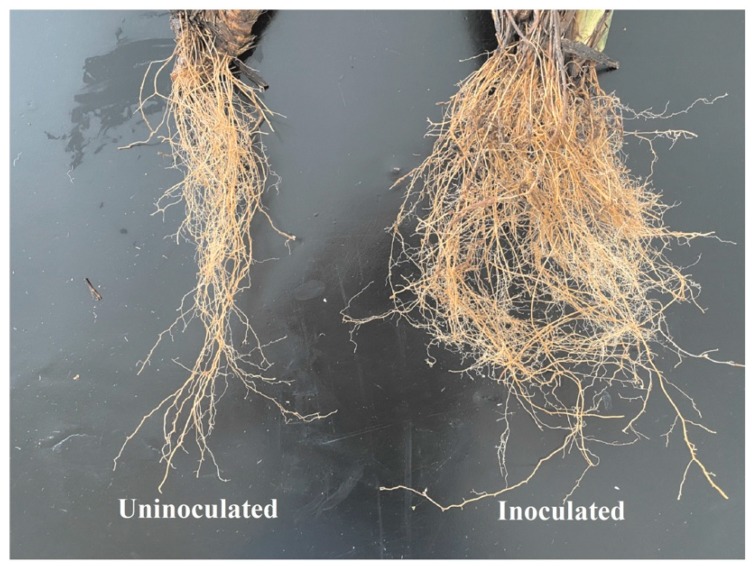
Root development of red rice grown under water restriction and inoculation conditions (50–55% field capacity).

**Table 1 ijms-21-00333-t001:** Quantification of indole-3-acetic acid (IAA), gibberellic acid 1 (GA_1_), gibberellic acid 3 (GA_3_), Cyt (trans-zeatin), trehalose (Treh), and α-tocopherol (α-toc) for the plants of red rice grown in the presence and absence of *G. diazotrophicus* strains Pal5, under (**a**) water restriction and (**b**) rehydrated conditions. The concentrations of different types of phytohormones were calculated in ng/mL, while trehalose and α-tocopherol were calculated in mg/L. Mean values and standard errors ( ± ) were calculated from the results of the three replicates. Averages followed by identical capital letters do not differ in the inoculation condition for each soil water treatment, and averages followed by identical lowercase letters do not differ in the percentage of soil water

Compounds	100%		70%		50%		30%	
	NI	I	NI	I	NI	I	NI	I
**(a)**								
IAA	4.21 ± 1.01 ^Bc^	7.37 ± 2.19 ^Ac^	10.22 ± 1.74 ^Bb^	33.92 ± 3.55 ^Ab^	15.29 ± 1.22 ^Ba^	360.23 ± 8.22 ^Aa^	2.77 ± 0.21 ^Ad^	3.1 ± 0.85 ^Ad^
GA_1_	6.44 ± 0.44 ^Bc^	8.77 ± 1.11 ^Ac^	10.74 ± 1.12 ^Bb^	22.41 ± 1.02 ^Ab^	12.11 ± 3.41 ^Ba^	81.44 ± 3.41 ^Aa^	2.28 ± 0.88 ^Ad^	3.21 ± 0.47 ^Ad^
GA_3_	8.17 ± 0.99 ^Bc^	9.11 ± 3.13 ^Ac^	11.12 ± 3.24 ^Bb^	37.19 ± 4.24 ^Ab^	13.54 ± 4.87 ^Ba^	90.77 ± 4.87 ^Aa^	5.74 ± 1.77 ^Ad^	5.07 ± 1.11 ^Ad^
Cyt	3.79 ± 0.21 ^Bc^	7.12 ± 1.91 ^Ac^	10.31 ± 2.01 ^Bb^	11.31 ± 2.11 ^Ab^	16.74 ± 4.74 ^Ba^	50.33 ± 4.34 ^Aa^	3.11 ± 0.11 ^Ad^	3.4 ± 0.98 ^Ad^
Treh	4.87 ± 0.44 ^Bc^	5.77 ± 0.87 ^Ac^	10.78 ± 2.41 ^Bb^	20.78 ± 2.11 ^Ab^	12.94 ± 3.22 ^Ba^	60.74 ± 3.44 ^Aa^	4.11 ± 0.77 ^Ad^	4.55 ± 0.85 ^Ad^
α-Toc	3.74 ± 0.41 ^Bc^	4.99 ± 0.47 ^Ac^	6.74 ± 0.74 ^Bb^	19.74 ± 0.44 ^Ab^	10.27 ± 2.94 ^Ba^	50.79 ± 2.77 ^Aa^	3.12 ± 0.84 ^Ad^	3.80 ± 0.81 ^Ad^
**(b)**								
IAA	3.99 ± 1.00 ^Ba^	6.37 ± 2.47 ^Aa^	5.74 ± 0.74 ^Aa^	6.44 ± 3.99 ^Aa^	6.89 ± 0.11 ^Aa^	7.10 ± 1.71 ^Aa^	6.33 ± 0.51 ^Aa^	5.1 ± 0.66 ^Aa^
GA_1_	5.82 ± 0.45 ^Aa^	5.77 ± 1.21 ^Aa^	5.76 ± 0.82 ^Aa^	5.73 ± 1.09 ^Aa^	6.55 ± 0.29 ^Aa^	7.23 ± 2.78 ^Aa^	6.79 ± 0.78 ^Aa^	5.88 ± 0.54 ^Aa^
GA_3_	9.11 ± 0.89 ^Aa^	9.78 ± 3.69 ^Aa^	8.54 ± 0.24 ^Aa^	10.33 ± 0.24 ^Aa^	11.45 ± 1.33 ^Aa^	10.33 ± 1.33 ^Aa^	10.1 ± 1.12 ^Aa^	9.89 ± 1.87 ^Aa^
Cyt	4.21 ± 0.74 ^Aa^	7.78 ± 1.96 ^Aa^	8.63 ± 0.71 ^Aa^	9.74 ± 0.21 ^Aa^	10.79 ± 2.44 ^Aa^	10.87 ± 0.87 ^Aa^	9.11 ± 0.12 ^Aa^	8.98 ± 0.25 ^Aa^
Treh	4.74 ± 0.55 ^Aa^	5.32 ± 0.58 ^Aa^	6.33 ± 0.41 ^Aa^	7.29 ± 0.31 ^Aa^	7.55 ± 2.22 ^Aa^	8.22 ± 0.55 ^Aa^	8.11 ± 0.45 ^Aa^	8.55 ± 0.87 ^Aa^
α-Toc	3.69 ± 0.75 ^Aa^	4.00 ± 0.47 ^Aa^	3.91 ± 0.75 ^Aa^	4.58 ± 0.82 ^Aa^	4.55 ± 0.33 ^Aa^	4.21 ± 1.22 ^Aa^	4.12 ± 0.78 ^Aa^	4.80 ± 0.85 ^Aa^

**Table 2 ijms-21-00333-t002:** RT-qPCR analysis of root growth and development of red rice genes during drought stress. Shown are the RT-qPCR results of the plants of red rice grown in the presence and absence of *G. diazotrophicus* strains Pal5, under water restriction. Mean values and standard errors were calculated from the results of the three replicates. Averages followed by identical capital letters do not differ in the inoculation condition for each soil water treatment, and averages followed by identical lowercase letters do not differ in the percentage of soil water. NI = Uninoculated and I = Inoculated.

Gene name	Gene ID	100%		70%		50%		30%	
		NI	I	NI	I	NI	I	NI	I
OsGNA1	Os09g0488000	1.0 ± 0.00	6.5 ± 0.11	5.5 ± 0.82	15.0 ± 0.55	9.3 ± 0.88	20.3 ± 0.74	7.2 ± 0.34	11.0 ± 0.11
OsMOGS	Os01g0921200	1.0 ± 0.00	9.3 ± 0.52	3.0 ± 0.36	17.3 ± 0.90	11.2 ± 0.12	29.9 ± 1.11	9.1 ± 0.88	10.2 ± 0.65
OsDGL1	Os07g0209000	1.0 ± 0.00	10.5 ± 1.11	7.0 ± 0.51	18.1 ± 0.60	13.0 ± 0.82	30.2 ± 0.23	4.0 ± 0.65	7.5 ± 0.44
OsCYT-INV1	Os02g0550600	1.0 ± 0.00	2.0 ± 0.43	2.5 ± 0.66	8.6 ± 0.33	4.8 ± 0.55	15.0 ± 0.66	1.0 ± 0.22	1.0 ± 0.92
OsGLR3.1	Os02g0117500	1.0 ± 0.00	17.0 ± 0.72	11.0 ± 0.39	22.4 ± 1.10	15.0 ± 0.13	40.7 ± 0.21	5.5 ± 0.92	2.1 ± 0.36
OsGatB	Os11g0544800	1.0 ± 0.00	8.0 ± 0.33	3.0 ± 0.11	14.0 ± 0.03	4.7 ± 0.54	28.0 ± 0.91	3.6 ± 0.43	2.9 ± 0.73
OsASL1	Os03g0305500	1.0 ± 0.00	22.5 ± 0.82	8.0 ± 0.81	30.9 ± 0.20	11.0 ± 0.62	50.0 ± 0.17	7.2 ± 0.16	3.0 ± 0.54
OsEL5	Os02g0559800	1.0 ± 0.00	17.0 ± 0.21	7.5 ± 0.21	28.0 ± 0.66	20.0 ± 0.11	45.0 ± 0.39	5.9 ± 0.22	2.3 ± 0.11
OsSPR1	Os01g0898300	1.0 ± 0.00	3.47 ± 0.88	6.5 ± 0.22	14.0 ± 0.59	9.9 ± 0.87	18.3 ± 0.73	4.2 ± 0.38	13.0 ± 0.21
OsMYB1	Os01g0128000	1.0 ± 0.00	3.33 ± 0.74	3.55 ± 0.76	20.3 ± 0.88	8.27 ± 0.18	39.9 ± 0.11	10.1 ± 0.31	20.2 ± 0.11
OsARF12	Os04g0671900	1.0 ± 0.00	11.5 ± 1.71	4.0 ± 0.59	15.1 ± 0.67	6.07 ± 0.77	20.2 ± 0.44	5.05 ± 0.45	6.5 ± 0.77
OsERF2	Os06g0181700	1.0 ± 0.00	4.07 ± 0.43	2.8 ± 0.86	9.6 ± 0.99	5.8 ± 0.74	17.0 ± 0.77	8.08 ± 0.78	9.08 ± 0.82
OsAIM1	Os02g0274100	1.0 ± 0.00	11.0 ± 0.45	8.77 ± 0.87	29.4 ± 0.10	11.00 ± 0.23	30.9 ± 0.91	8.5 ± 0.82	8.1 ± 0.88
SHB	Os05g0389000	1.0 ± 0.00	10.0 ± 0.11	2.0 ± 0.41	18.0 ± 0.47	4.1 ± 0.11	22.0 ± 0.41	4.6 ± 0.43	4.9 ± 0.44
ARL1/CRL1	Os03g0149100	1.0 ± 0.00	4.54 ± 0.44	3.03 ± 0.43	10.9 ± 0.44	6.04 ± 0.46	34.0 ± 0.87	7.8 ± 0.17	8.0 ± 0.57
OsIAA23	Os06g0597000	1.0 ± 0.00	5.0 ± 0.87	2.57 ± 0.21	2.00 ± 0.66	1.89 ± 0.11	1.55 ± 0.39	1.21 ± 0.22	1.11 ± 0.11
OsNAL1	Os04g0615000	1.0 ± 0.00	5.5 ± 0.81	2.5 ± 0.88	10.0 ± 0.44	7.3 ± 0.71	15.3 ± 0.49	7.2 ± 0.55	10.0 ± 0.01
OsCHR4/CRL6	Os07g0497100	1.0 ± 0.00	20.2 ± 0.72	7.0 ± 0.88	28.3 ± 0.97	10.2 ± 0.72	39.9 ± 0.11	9.11 ± 0.18	10.4 ± 0.45
OsCKX4	Os01g0940000	1.0 ± 0.00	15.5 ± 0.11	4.04 ± 0.41	28.1 ± 0.20	6.06 ± 0.42	30.4 ± 0.24	4.6 ± 0.75	5.5 ± 0.54
OsIAA11	Os03g0633500	1.0 ± 0.00	8.0 ± 0.93	7.5 ± 0.86	6.6 ± 0.23	6.8 ± 0.75	6.00 ± 0.96	1.0 ± 0.22	1.2 ± 0.92
OsSLL1	Os04g0379900	1.0 ± 0.00	11.0 ± 0.72	4.09 ± 0.78	20.4 ± 0.10	5.0 ± 0.43	40.9 ± 0.91	4.5 ± 0.12	5.1 ± 0.56
OsORC3	Os10g0402200	1.0 ± 0.00	9.0 ± 0.33	4.8 ± 0.11	18.0 ± 0.07	6.7 ± 0.54	23.0 ± 0.74	4.6 ± 0.87	5.9 ± 0.79
OsHOX1	Os10g0561800	1.0 ± 0.00	3.33 ± 0.22	1.5 ± 0.81	9.99 ± 0.29	5.05 ± 0.25	24.0 ± 0.44	4.2 ± 0.15	5.0 ± 0.55

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
