# Peer review of "Gluconacetobacter diazotrophicus* Changes The Molecular Mechanisms of Root Development in *Oryza sativa* L. Growing Under Water Stress"

_ijms, 2020, doi:10.3390/ijms21010333_

Round 1
Reviewer 1 Report
In this manuscript, Silva, Meneses and collaborators describe the effect of drought and inoculation with G. diazotrophicus, as well rehydration of red rice on growth, photosynthesis, hormone levels, and expression of genes involved in plant development. Overall, the study provides interesting results regarding the positive effect of bacterial inoculation on plant growth under stress conditions. Interestingly, a large set of data was collected that provides a wide picture of the plant status under these conditions.
The paper, however, must be improved in terms of writing since some grammar and syntax errors are present in the manuscript. Also, a better description of the results should be considered (see below). Furthermore, the discussion section must be more focused on your own data. For instance, explain how all the variables that were measured correlate among each other in light of the observed results? How do you explain that the positive effect of the microbes is only observed under stress? etc. I kindly suggest reorganizing and shortening this section.
Also, this study might be strengthened by an analysis of multiple correlations, and also a principal component analysis to understand which variables explain most of the observations.
Below, I present some specific comments.
Line 18 “Gluconacetobacter diazotrophicus associated with red rice increase plant root development; however, recent work by several groups has shown that G. diazotrophicus also elicit…” This sentence is not clear. Maybe you mean “Inoculation with Gluconacetobacter diazotrophicus has shown to influence root development in red rice plants, and more recently to also elicit the induced systemic tolerance response to drought”.
Line 20 “This work aimed to” -> “the goal of this study was to”.
Line 22 “The experiment treatments of the red rice cultivar inoculated and uninoculated with G. diazotrophicus, and cultivated with and without water restriction.” This sentence is not clear.
Line 37 “Reports indicated water deficiency…” please review this sentence
Line 40 You mean “Plant growth-promoting bacteria”
Line 61 “…owing to its the responses of roots to drought and their role under drought conditions...” Review this whole sentence.
Line 71 qPCR by itself does not allow to look at gene expression, maybe you mean RT-qPCR.
Line 77 Different time points or different water availability levels?
Line 80 Although four different parameters were measured (i.e. shoot length, leaf area, root length, and lateral roots), the results seem to be focused solely on shoot length. Could you please describe the effect of both inoculation and water availability on the other parameters quantified?
Line 81 What do you mean with “…levels of water restriction on the accumulation of growth on shoots…”, maybe you mean “levels of water restriction on growth on shoots…”
Line 83 “The water restriction…” better “Water restriction”. And “…affected the growth…”
Line 106 Maybe you mean Fig. 2A.
Line 107 You may describe what is the purpose of rehydration, and why this is important. Also, briefly mention what is the definition of Efficiency of Water Usage and Instantaneous Efficiency of Carboxylation, and why this is relevant. The description of these concepts will improve the flow of the manuscript.
Line 119 Same as above. You might start with: “The initial fluorescence (F0) value describes is a measure of the steady-state fluorescence signal, which is correlated with XXX. To determine the impact of drought and inoculation on F0, we measured this parameter in untreated and treated plants growing at different water availability percentages. Results showed a decrease in … Next, we explored the effect of rehydration, and observed that…”
Line 131 Same as above.
Line 140 “The same behavior was not observed for uninoculated plants.” Could you please review this affirmation? Results showed that drought negatively impacted the level of pigments regardless of bacterial inoculation; yet, under these conditions of water limitation, inoculation always improved the content of photosynthetic pigments.
Line 155 Please review this sentence “The analytes were found both in the plants cells compared to the control” maybe you want to say something like “Overall, a decrease in water availability and inoculation positively correlated with an increase in hormones concentration. This suggests that the hormones are secreted as part of the stress response, and also that bacterial inoculation can directly and/or indirectly affect hormone plant levels.”
Line 205 It is very interesting to see how, in the absence of stress, inoculation leads to increased expression of the measured genes. This result provides evidence of a direct effect of the microbe on plant development. How can you explain the fact that there is increased expression of the genes, yet no differences in plant size?
Line 250 See above
Line 463 Is the inoculation interesting because the PGPB are being more widely used? Or because they represent an agronomic alternative to increase yield in a sustainable manner?
Author Response
Dear Reviewer,
The constructive criticisms of referees certainly contributed to the improvement of the document. We attended most of the points raised by the reviewers, modified others and addressed detailed answers to those that we do not agree with them. An editorial office in USA (EDITAGE - Cactus Communications) edited the document, therefore its quality has improved a lot and should be at a standard level of the IJMS.
Reviewer 1
In this manuscript, Silva, Meneses and collaborators describe the effect of drought and inoculation with G. diazotrophicus, as well rehydration of red rice on growth, photosynthesis, hormone levels, and expression of genes involved in plant development. Overall, the study provides interesting results regarding the positive effect of bacterial inoculation on plant growth under stress conditions. Interestingly, a large set of data was collected that provides a wide picture of the plant status under these conditions. Thanks
The paper, however, must be improved in terms of writing since some grammar and syntax errors are present in the manuscript. Also, a better description of the results should be considered (see below). Furthermore, the discussion section must be more focused on your own data. For instance, explain how all the variables that were measured correlate among each other in light of the observed results? How do you explain that the positive effect of the microbes is only observed under stress? etc. I kindly suggest reorganizing and shortening this section. Suggestion answered
Also, this study might be strengthened by an analysis of multiple correlations, and also a principal component analysis to understand which variables explain most of the observations. Suggestion answered
Below, I present some specific comments.
Line 18 “Gluconacetobacter diazotrophicus associated with red rice increase plant root development; however, recent work by several groups has shown that G. diazotrophicus also elicit…” This sentence is not clear. Maybe you mean “Inoculation with Gluconacetobacter diazotrophicus has shown to influence root development in red rice plants, and more recently to also elicit the induced systemic tolerance response to drought”. Suggestion answered
Line 20 “This work aimed to” -> “the goal of this study was to”. Suggestion answered
Line 22 “The experiment treatments of the red rice cultivar inoculated and uninoculated with G. diazotrophicus, and cultivated with and without water restriction.” This sentence is not clear. Clarified sentence in the manuscript
Line 37 “Reports indicated water deficiency…” please review this sentence Clarified sentence in the manuscript
Line 40 You mean “Plant growth-promoting bacteria” Suggestion answered
Line 61 “…owing to its the responses of roots to drought and their role under drought conditions...” Review this whole sentence. Clarified sentence in the manuscript
Line 71 qPCR by itself does not allow to look at gene expression, maybe you mean RT-qPCR. Suggestion answered
Line 77 Different time points or different water availability levels? Different water availability levels, suggestion answered
Line 80 Although four different parameters were measured (i.e. shoot length, leaf area, root length, and lateral roots), the results seem to be focused solely on shoot length. Could you please describe the effect of both inoculation and water availability on the other parameters quantified? Suggestion answered
Line 81 What do you mean with “…levels of water restriction on the accumulation of growth on shoots…”, maybe you mean “levels of water restriction on growth on shoots…” Suggestion answered
Line 83 “The water restriction…” better “Water restriction”. And “…affected the growth…” Suggestion answered
Line 106 Maybe you mean Fig. 2A. Suggestion answered
Line 107 You may describe what is the purpose of rehydration, and why this is important. Also, briefly mention what is the definition of Efficiency of Water Usage and Instantaneous Efficiency of Carboxylation, and why this is relevant. The description of these concepts will improve the flow of the manuscript. Suggestion answered
Line 119 Same as above. You might start with: “The initial fluorescence (F0) value describes is a measure of the steady-state fluorescence signal, which is correlated with XXX. To determine the impact of drought and inoculation on F0, we measured this parameter in untreated and treated plants growing at different water availability percentages. Results showed a decrease in … Next, we explored the effect of rehydration, and observed that…” Suggestion answered
Line 131 Same as above. Suggestion answered
Line 140 “The same behavior was not observed for uninoculated plants.” Could you please review this affirmation? Results showed that drought negatively impacted the level of pigments regardless of bacterial inoculation; yet, under these conditions of water limitation, inoculation always improved the content of photosynthetic pigments. Clarified sentence in the manuscript
Line 155 Please review this sentence “The analytes were found both in the plants cells compared to the control” maybe you want to say something like “Overall, a decrease in water availability and inoculation positively correlated with an increase in hormones concentration. This suggests that the hormones are secreted as part of the stress response, and also that bacterial inoculation can directly and/or indirectly affect hormone plant levels.” Suggestion answered
Line 205 It is very interesting to see how, in the absence of stress, inoculation leads to increased expression of the measured genes. This result provides evidence of a direct effect of the microbe on plant development. How can you explain the fact that there is increased expression of the genes, yet no differences in plant size? We can explain this phenomenon in the following ways: Although gene expression is increased in inoculated plants (100% of water availability) this does not mean a phenotypic expression with the same tendency, because gene expression occurs almost immediately and strongly, contrary to expression phenotypic. Another factor that can be observed is that despite the increased gene expression, phytohormones values are slowly following this tendency, as shown in table 1a (100% of water availability). Another factor found is that we observed a small growth of inoculated plants (100% of water availability), but this is not statistically significant. Finally, we observed that an increase occurred mainly in the auxin-producing genes (IAA) figure 5a and b, the action of auxins is clearly effected in root systems, so we did not expect an increase in plant size.
Line 250 See above We can explain this phenomenon in the following ways: Although gene expression is increased in inoculated plants (100% of water availability) this does not mean a phenotypic expression with the same tendency, because gene expression occurs almost immediately and strongly, contrary to expression phenotypic. Another factor that can be observed is that despite the increased gene expression, phytohormones values are slowly following this tendency, as shown in table 1a (100% of water availability). Another factor found is that we observed a small growth of inoculated plants (100% of water availability), but this is not statistically significant. Finally, we observed that an increase occurred mainly in the auxin-producing genes (IAA) figure 5a and b, the action of auxins is clearly effected in root systems, so we did not expect an increase in plant size.
Line 463 Is the inoculation interesting because the PGPB are being more widely used? Or because they represent an agronomic alternative to increase yield in a sustainable manner? Suggestion answered
Best Regards
Reviewer 2 Report
In this manuscripts, authors investigated the effects of G. diazotrophicus Pal5 on development and responses of red rice under drought stress, mainly focusing on root development. Although association between plant hormone levels, gene expression and red rice root development were detected, further investigation is needed to elucidate the mechanisms that Pal5 affects red rice root development, e.g. whether Pal5 secret these hormones, and how Pal5 affects plant hormone production.
Certain results of this manuscript and results of Filgueiras et al. 2019. (Plant Soil doi:10.1007/s11104-019-04163-1) are redundant. Filgueiras et al. found that inoculation with Gluconacetobacter diazotrophicus Pal5 played positive roles on red rice cultivar under water stress, including plant development, gas exchange, osmoprotectant solutes in shoots, seed yield, antioxidant enzyme activities, and defense genes. Although different parameters were used in this manuscript, e.g. “2.1. Plant development” and “2.2. Gas exchange”, the conclusion is the same as Filgueiras et al. 2019.
Although the figures and tables in this manuscript were clear and well-organized, the description of the data in the Results part was under satisfaction. Briefly, most of the description of data is too general and vague to be understood easily or to be accurate. Also, some of the results are plain description/list of numbers, without rationale and conclusion.
For example,
Line 83-85: “The water restriction of up to 50% affected the growth and development in both presence and absence of bacterial inoculation, and did not differ from the uninoculated plants grown at 100% field capacity.” Which data does this sentence refer to? Fig. 1a, 1b and 1c did show significant difference in both presence and absence of bacteria at 50% compared to 100% water availability. However, in Fig 1d, without inoculation, lateral roots was similar between 50% and 100% water availability.
Line 86-88: “In contrast, water restriction at 70% promote higher growth, similar to the inoculated plants at 100% field capacity. The growth of inoculated plants grown at water restriction level of 70% did not statistically differ from the inoculated plants grown at 100% field capacity (control)”. What does “higher growth” refer to? Only Fig 1a showed similar shoot length of inoculated plants between 70% and 100% water availability. For other panels, significant differences were detected in inoculated plants between 70% and 100% water availability.
Line 106: “inoculation with G. diazotrophicus promoted higher Efficiency of Water Usage (Figure a) and Instantaneous Efficiency of Carboxylation (Figure 2b)”. This is not true if comparison was made between “uninoculated rehydrated” and “inoculated rehydrated”. At water availability of 70%, 50% and 30%, comparing among “uninoculated”, “inoculated”, “uninoculated rehydrated” and “inoculated rehydrated”, inoculation alleviated the suppression of Efficiency of Water Usage and Instantaneous Efficiency of Carboxylation with water stress, other than promote higher efficiency.
Line 122-124: “Under hydration effect, it was found that there was an increase in F0 concentration in relation to the stress period, but there was no significant difference between the 100% levels”. At 70% water availability, there was no significant difference between “Inoculated” and “Uninoculated rehydrated”/“Inoculated rehydrated”.
Line 154-155: “the other phytohormones profiled”, what are the other phytohormones profiled?
More inaccurate or vague description was listed as follows: Line 131-135, Line 196-199, Line 211-212
Please provide more detailed description and specify each comparison (water availability, inoculated or uninoculated, rehydrated or not) when describing suppression or enhancement.
Moreover, the Results part should also be refined:
1, please add at the beginning of each section to explain why this data was collected and which aspect of plant response this data indicated.
For example, “2.4. Phytohormones and osmoprotective solutes”, what are IAA, GAs, Cyt, trehalose and α-tocopherol, and why they were investigated? Of which aspect of plant responses, these compounds could give us insights?
2, at the end of each section of results, please add a summary/conclusion.
For example, “2.4. Phytohormones and osmoprotective solutes”, what does the data of hormones and osmoprotective solutes indicate?
“2.4. qPCR analysis of the root growth and development genes”, “two genes had their expression levels decreased”. Why these two genes different from the other 21? Negative regulator? What does this indicate?
3, for 2.1. Plant development, please describe each panel of Figure 1, separately. Each panel in Figure 1 indicates different aspects of plant growth, and the trends between certain panels are different from each other, e.g. inoculation alleviate the suppression of shoot growth at 50% water availability compared to no inoculation (Fig 1a), while inoculation enhanced the induction of root growth at 50% water availability compared to no inoculation (Fig 1c). Otherwise, the results are hard to understand and might lead to confusion.
Line 155-157: “The analytes were found both in the plants cells compared to the control, indicating G. diazotrophicus strains Pal5 may secrete both compounds, as well as estimate the plants to produce them”. This conclusion can not be reached based on Table 1. Is there any data showed that Pal5 secrete these compounds?
Minor:
1, Line 416-418 indicated that trehalose and α-tocopherol were collected from roots, while Line 27-28 “Increase in root area and higher levels of gas exchange and osmoprotectant solutes were observed in shoots”. Please verify.
2, Figure legend of Figure 4 is not correct, which is a copy of the legend of Figure 3.
3, Line 76-77: “red rice plants were collected at different time points of various drought stresses”. No different time points were mentioned in the Results part. Please specify.
4, Incomplete sentences: Line 143-144, “photosynthetic, this behavior for both G. diazotrophicus inoculated and uninoculated plants.”
5, Grammatically: Line 259-260, “Bacteria that produce growth hormones like Indole acetic acid and which display Gibberellic acid activity can help plants tolerate water stress by increasing the root growth and by reducing the water stress perception in plants.”
6, Unidentified abbreviation:
Line 132: “CC”
Line 133: “WC”
7, Typo:
Line 105: “gas exchance”
Line 106: “Figure a”
Line 205: “2.4. qPCR analysis of the root growth and development genes”
Author Response
Dear Reviewer,
The constructive criticisms of referees certainly contributed to the improvement of the document. We attended most of the points raised by the reviewers, modified others and addressed detailed answers to those that we do not agree with them. An editorial office in USA (EDITAGE - Cactus Communications) edited the document, therefore its quality has improved a lot and should be at a standard level of the IJMS.
Reviewer 2
In this manuscripts, authors investigated the effects of G. diazotrophicus Pal5 on development and responses of red rice under drought stress, mainly focusing on root development. Although association between plant hormone levels, gene expression and red rice root development were detected, further investigation is needed to elucidate the mechanisms that Pal5 affects red rice root development, e.g. whether Pal5 secret these hormones, and how Pal5 affects plant hormone production. Clarified sentence in the manuscript
Certain results of this manuscript and results of Filgueiras et al. 2019. (Plant Soildoi:10.1007/s11104-019-04163-1) are redundant. Filgueiras et al. found that inoculation with Gluconacetobacter diazotrophicus Pal5 played positive roles on red rice cultivar under water stress, including plant development, gas exchange, osmoprotectant solutes in shoots, seed yield, antioxidant enzyme activities, and defense genes. Although different parameters were used in this manuscript, e.g. “2.1. Plant development” and “2.2. Gas exchange”, the conclusion is the same as Filgueiras et al. 2019. Clarified sentence in the manuscript
Although the figures and tables in this manuscript were clear and well-organized, the description of the data in the Results part was under satisfaction. Briefly, most of the description of data is too general and vague to be understood easily or to be accurate. Also, some of the results are plain description/list of numbers, without rationale and conclusion. Suggestion answered
For example,
Line 83-85: “The water restriction of up to 50% affected the growth and development in both presence and absence of bacterial inoculation, and did not differ from the uninoculated plants grown at 100% field capacity.” Which data does this sentence refer to? Fig. 1a, 1b and 1c did show significant difference in both presence and absence of bacteria at 50% compared to 100% water availability. However, in Fig 1d, without inoculation, lateral roots was similar between 50% and 100% water availability. Suggestion answered
Line 86-88: “In contrast, water restriction at 70% promote higher growth, similar to the inoculated plants at 100% field capacity. The growth of inoculated plants grown at water restriction level of 70% did not statistically differ from the inoculated plants grown at 100% field capacity (control)”. What does “higher growth” refer to? Only Fig 1a showed similar shoot length of inoculated plants between 70% and 100% water availability. For other panels, significant differences were detected in inoculated plants between 70% and 100% water availability. Suggestion answered
Line 106: “inoculation with G. diazotrophicus promoted higher Efficiency of Water Usage (Figure a) and Instantaneous Efficiency of Carboxylation (Figure 2b)”. This is not true if comparison was made between “uninoculated rehydrated” and “inoculated rehydrated”. At water availability of 70%, 50% and 30%, comparing among “uninoculated”, “inoculated”, “uninoculated rehydrated” and “inoculated rehydrated”, inoculation alleviated the suppression of Efficiency of Water Usage and Instantaneous Efficiency of Carboxylation with water stress, other than promote higher efficiency. Suggestion answered
Line 122-124: “Under hydration effect, it was found that there was an increase in F0 concentration in relation to the stress period, but there was no significant difference between the 100% levels”. At 70% water availability, there was no significant difference between “Inoculated” and “Uninoculated rehydrated”/“Inoculated rehydrated”. Suggestion answered
Line 154-155: “the other phytohormones profiled”, what are the other phytohormones profiled? Clarified sentence in the manuscript
More inaccurate or vague description was listed as follows: Line 131-135, Line 196-199, Line 211-212 Suggestion answered and clarified sentence in the manuscript
Please provide more detailed description and specify each comparison (water availability, inoculated or uninoculated, rehydrated or not) when describing suppression or enhancement. Suggestion answered
Moreover, the Results part should also be refined:
1, please add at the beginning of each section to explain why this data was collected and which aspect of plant response this data indicated. Suggestion answered
For example, “2.4. Phytohormones and osmoprotective solutes”, what are IAA, GAs, Cyt, trehalose and α-tocopherol, and why they were investigated? Of which aspect of plant responses, these compounds could give us insights? Suggestion answered and clarified sentence in the manuscript
2, at the end of each section of results, please add a summary/conclusion. Suggestion answered
For example, “2.4. Phytohormones and osmoprotective solutes”, what does the data of hormones and osmoprotective solutes indicate? Suggestion answered
“2.4. qPCR analysis of the root growth and development genes”, “two genes had their expression levels decreased”. Why these two genes different from the other 21? Negative regulator? What does this indicate? Suggestion answered and clarified sentence in the manuscript
3, for 2.1. Plant development, please describe each panel of Figure 1, separately. Each panel in Figure 1 indicates different aspects of plant growth, and the trends between certain panels are different from each other, e.g. inoculation alleviate the suppression of shoot growth at 50% water availability compared to no inoculation (Fig 1a), while inoculation enhanced the induction of root growth at 50% water availability compared to no inoculation (Fig 1c). Otherwise, the results are hard to understand and might lead to confusion. Suggestion answered and clarified sentence in the manuscript
Line 155-157: “The analytes were found both in the plants cells compared to the control, indicating G. diazotrophicus strains Pal5 may secrete both compounds, as well as estimate the plants to produce them”. This conclusion can not be reached based on Table 1. Is there any data showed that Pal5 secrete these compounds? Suggestion answered and clarified sentence in the manuscript. “The phytohormones were found both in the…”
Minor:
1, Line 416-418 indicated that trehalose and α-tocopherol were collected from roots, while Line 27-28 “Increase in root area and higher levels of gas exchange and osmoprotectant solutes were observed in shoots”. Please verify. Suggestion answered and clarified sentence in the manuscript
2, Figure legend of Figure 4 is not correct, which is a copy of the legend of Figure 3. Suggestion answered and clarified sentence in the manuscript
3, Line 76-77: “red rice plants were collected at different time points of various drought stresses”. No different time points were mentioned in the Results part. Please specify. Suggestion answered and clarified sentence in the manuscript
4, Incomplete sentences: Line 143-144, “photosynthetic, this behavior for both G. diazotrophicus inoculated and uninoculated plants.” Suggestion answered and clarified sentence in the manuscript
5, Grammatically: Line 259-260, “Bacteria that produce growth hormones like Indole acetic acid and which display Gibberellic acid activity can help plants tolerate water stress by increasing the root growth and by reducing the water stress perception in plants.” Suggestion answered and clarified sentence in the manuscript
6, Unidentified abbreviation:
Line 132: “CC” Suggestion answered and clarified sentence in the manuscript
Line 133: “WC” Suggestion answered and clarified sentence in the manuscript
7, Typo:
Line 105: “gas exchance” Suggestion answered
Line 106: “Figure a” Suggestion answered
Line 205: “2.4. qPCR analysis of the root growth and development genes” Suggestion answered
Round 2
Reviewer 2 Report
The authors have successfully addressed the reviewer’s comments. And the revised manuscript has been significantly improved, which is now more precise and intriguing. The authors also provided evidence that Pal5 might regulate red rice root development through mediating plant hormone production as well as through synthesizing bacterial hormones by itself. Hence, this manuscript conveys potentially important findings and should be of interest to a broad audience.
Author Response
Dear Review,
We appreciate the compliments for the quality of the work and the interest in our results.
Best Regards